# Carbogen gas-challenge BOLD fMRI in assessment of liver hypoxia after portal microcapsules implantation

**Yuefu Zhan[1], Yehua Wu[2], Jianqiang Chen[3]***

**1** Department of Radiology, Maternal and Child Health Hospital of Hainan Province, Haikou, Hainan, China, **2** Hainan General Hospital, Haikou, China, **3** Department of Radiology, Xiangya School of Medicine Affiliated Haikou Hospital, Central South University, Haikou, Hainan, China

* hnchenjq@163.com

**Data Availability Statement:** All relevant data are within the manuscript and deposited to Figshare (https://doi.org/10.6084/m9.figshare.10302212.v1).

## Abstract

### Background

Hypoxia is one of the key factors affecting the survival of islet cells transplanted via the portal vein. Blood oxygen level dependent functional magnetic resonance imaging (BOLD-fMRI) is the only imaging technique that can detect the level of blood oxygen level in vivo. However, so far no study has indicated that BOLD-fMRI can be applied to monitor the liver oxygen level after islet transplantation.

### Objective

To evaluate the value of Carbogen-challenge BOLD MRI in assessing the level of hypoxia in liver tissue after portal microcapsules implanted.

### Methods

Fifty-one New Zealand rabbits were randomly divided into three experimental groups (15 in each group) were transplanted microencapsulated 1000 microbeads/kg (PV1 group), 3000 microbeads/kg (PV2 group), 5000 microbeads/kg (PV3 group), and 6 rabbits were injected with the same amount of saline as the control group, BOLD-fMRI was performed following carbogen breathing in each group after transplantation on 1d, 2d, 3d and 7d, T2* weighted image, R2* value and ΔR2* value parameters for the liver tissue. Pathological examinations including liver gross pathology, H&E staining and pimonidazole immunohistochemistry were performed after BOLD-fMRI. The differences of pathological results among each group were compared. The ΔR2* values and transplanted doses were analyzed.

### Results and conclusions

ΔR2* values at the 1-3d and 7d after transplantation were significantly different in each groups (P<0.05). ΔR2* values decreased gradually with the increase of transplanted dose, and was negatively correlated with transplant dose at 3d after transplantation (r = -0.929, P <0.001). Liver histopathological examination showed that the degree of hypoxia of liver

**Funding:** This work was supported by Hainan Province Scientific and Technological Program (No. ZDYD2019164) and the Health Family Planning Project in Hainan Province (16A200087). The funders had no role in study design, data collection and analysis, decision to publish, or preparation of the manuscript.

**Competing interests:** The authors have declared that no competing interests exist.

tissue increased with the increase of transplanted doses, Carbogen-challenge BOLD-fMRI can assess the degree of liver hypoxia after portal microcapsules implanted, which provided a monitoring method for early intervention.

## Introduction

Islet cell transplantation is an ideal treatment of type I diabetes mellitus and can effectively prevent the progress of diabetes mellitus [1–3]. However, due to the immunological rejection after transplantation leading to impaired islet graft function, patients often have to re-take the exogenous insulin or re-transplant [4–5]. Microencapsulated pancreatic islet cells with translucent membrane provide a possibility to solve immunological rejection [6–8]. However, the long-term follow-up results are poor due to hypoxia and other problems. The rate of patients who do not rely on exogenous insulin within 5 years after transplantation is only 8.5% [9–11].

It is believed that the survival of islet cells after transplantation depends mainly on the revascularization of blood supply, which originates from the host. The vascular system of the host promotes the regeneration of blood vessels, while the endothelial cells retained when the islet cells are separated participate in the process of revascularization. However, the revascularization will not begin immediately. It usually starts about 2–4 days after the operation and takes about 10–14 days to complete. Before the establishment of blood supply, islet cells mainly depended on oxygen permeation with surrounding liver parenchyma to maintain their function and survival. In order to achieve the therapeutic effect of diabetes mellitus, a large amount of microcapsules transplantation is needed. Large dose of microcapsules will stay in the small branches of portal vein after entering the liver, which may cause vascular embolism, leading to ischemic blood oxygen in the surrounding liver tissue, necrosis and apoptosis, seriously affecting the oxygen permeation of islet cells, thereby leading to islet cell damage, and impairing the liver function, affecting the therapeutic effect [12–15]. In order to alleviate the degree of hypoxia and hepatic vascular embolism, the main method currently used is to undergo multiple rounds of transplantations. However, with the increase of transplantation times, the probability of recipient infection is rising, which also greatly increases the cost of transplantation and economic burden for those patients. Therefore, how to solve the above contradictions according to individual differences requires a better grasp of the relationship between transplantation volume and frequency, and a quantitative detection of target hypoxia after transplantation. Moreover, the detection of hypoxia in the target area after transplantation can reflect the effectiveness of immune activation in the early stage of transplantation, the recovery level of hepatic hemodynamics, and provide relevant evidence for clinical intervention.

At present, the monitoring method of liver hypoxia level in transplantation area is limited to target tissue biopsy. This invasive method not only increases the infection rate but also the pain of recipients. Therefore, a safe, non-invasive, real-time, dynamic, effective and simple method is urgently needed to monitor the hypoxia level of liver after transplantation.

Blood oxygen level-dependent functional magnetic resonance imaging (BOLD-fMRI), acting via modulation of the $T2^*$-weighted signal by changes in the ratio of blood paramagnetic deoxyhemoglobin to diamagnetic oxyhemoglobin, is the only technique that can detect the level of blood oxygen in vivo. It is widely used in the fields of central nervous system and liver [16–19]. By observing the changes of $T2^*$ signal, we can detect the changes of blood oxygen content in local tissues and judge the degree of hypoxia and hemodynamic changes. In this study, we simulated the clinical islet transplantation process with different number of

microcapsules after portal vein transplantation in rabbits, and performed BOLD-fMRI by Carbogen administration (95% O2 and 5% CO2) at different time points after operation. The purpose of this study was to explore the relationship between $\Delta R2^*$ value of liver before and after stimulation and liver hypoxia level after different dosages of microcapsules transplantation, and to further discuss the BOLD-fMRI in evaluating the hypoxia level of liver tissue after portal vein microcapsule transplantation.

## Materials and methods

### Animals and groups

The study was approved by the Ethical Review Committees (which is equivalent to IACUC) of the Xiangya school of medicine affiliated Haikou hospital, Central South University. We can confirm that all methods were performed in accordance with the guidelines for the care and use of experimental animals developed by the Chinese Society of Laboratory Animal Science.

Forty-one adult males and ten adult females New Zealand white rabbits (6 months old with 2.0–3.1kg body weight) were used in this experiment (provided by Animal Laboratory Center of the Third Affiliated Hospital of Xiangya Medical College, Central South University): (SCXK (Hunan) 2014/0011). The animals were raised under SPF (Specific pathogen free) condition with a 12h light-dark schedule. All animals had free access to food and water. They were randomly divided into three groups: PV1, PV2 and PV3 (fifteen animals in each group). In each group, the animals were transplanted with microcapsules of 1000 microbeads/kg, 3000 microbeads/kg and 5000 microbeads/kg, respectively. Six animals were injected with the same amount of saline as control.

### Experimental method

Microcapsules were prepared in 1.5% sodium alginate solution, and then dripping into CaCl2 solution to form calcium alginate microspheres. Then they were mixed with 0.05% poly L-lysine solution, 0.15% sodium alginate solution and sodium citrate solution, respectively. The diameter of the microcapsules was about 109±26μm. The morphology of microcapsules was observed by optical microscopy. 10 ml of microcapsule suspension was counted under optical microscopy.

### Portal vein microcapsule transplantation

After anesthesia with 3% pentobarbital sodium (1.0ml/kg) by intravenous injection into ear margin, the animals were fixed on the operating table in supine position. A 4-5cm incision was made. The portal vein was readily detected in the liver lobe. The microcapsules were injected slowly in the portal vein for 10-15s with a total amount of 1ml. In the control group, 1 ml saline was injected into the portal vein, and the procedure was the same as the above.

After the surgery, the animals were raised individually and treated with dexamethasone and polymyxin twice a day on the incision to prevent from infection. Every day the staff of the experimental animal center made a careful examination of the animals based on the appearance and behavior. No animal died unexpectedly before the experiments.

### MRI scanning

The animals were generally anesthetized with 3% sodium pentobarbital (1.0ml/kg) by intravenous injection at auricular margin. All animals were scanned by MRI (Siemens Avanto 1.5T) on the 1st, 2nd, 3rd and 7th day after operation, fasting for 12 hours before scanning. Scanning sequence included transverse T1WI, transverse and coronal fat suppression T2WI, transverse

BOLD. BOLD scanning parameters are: TR 150 ms, TE 3.4–39.2 ms, 5.0 mm sacnning thickness with 2.0mm intervals, matrix 192*78 mm, FA 30 degrees, FOV 150*60 mm2.

Air and Carbogen gas were inhaled through the mask, respectively. The first BOLD scan of the liver was performed after 10 minutes of inhalation of air, and then repeat the scan after 10 minutes of Carbogen gas inhalation. The BOLD scan was performed at a flow rate of 15 ml/s.

### Image analysis and post-processing

After BOLD image scanning, the original T2* image and pseudo-color image were automatically generated. Five regions of interest were selected from the left lateral lobe, left inner lobe, right outer lobe, right inner lobe and caudate lobe of five layers near the hepatic hilum, with a total size of about 25–45 voxels. The artifacts of intrahepatic blood vessels, bile ducts and chemical displacement were avoided, and the same regions of interest were ensured before and after inhalation of air and Carbogen gas. Mean R2* value (1/T2*), and then calculate $\Delta$R2* value, $\Delta$R2* value = R2* air-R2* Carbogen gas.

### Histopathological examination

Three animals in each experimental group were sacrificed by injection of overdosed 3% sodium pentobarbital (2.0ml/kg) at the end of 1, 2, 3 and 7 days after operation. The control group (6 animals) was sacrificed at 7 days after operation. Pimonidazole, a hypoxia marker, was injected into the abdominal cavity of the animals at a dose of 60 mg/kg 2 hours before execution. The rabbit liver tissue was taken within 30 minutes after execution. Paraffin sections were stained with HE and pimonidazole immunohistochemistry. The region was the same as the area of interest scanned in BOLD-fMRI. Immunohistochemical staining criteria: brown granules in the cytoplasm were positive cells. Five ROIs were randomly selected under microscopy (100X), and 100 hepatocytes were counted in each ROI (totally 500 cells). The data were analyzed with the method proposed by Nordsmark et al. [20]. Briefly, percentage of positive cells: positive cells accounted for less than 5%, 6%-15%, 16% - 30% and more than 30% of all cells were assigned with 1, 2,3 and 4 score, respectively. Positive staining intensity: 1 score for non-staining, 2 score for light staining, 3 score for moderate and 4 score for heavy staining, respectively. Hypoxia level = Percentage of positive cells * intensity of positive staining. If the score is in 0–3, then marked as (-). If in 4–6, 7–9, 10–12 then marked as (+), (++),(+++), respectively.

### Statistical analysis

All the data were analyzed with SPSS 17.0 statistical software. The correlation between the$\Delta$R2* value and the transplantation dose was analyzed by non-parametric Spearman correlation analysis, P < 0.05 was statistically significant.

## Results

### Microcapsules transplantation induced defect in liver

In control group, the liver surface and parenchyma were normal; in PV1 group, the color of liver surface was normal 3 days after operation, HE staining showed enlargement of hepatic sinuses around portal vein and no degeneration and necrosis of hepatocytes; in PV2 group, a few patchy white ischemic foci appeared on the liver surface, HE staining showed enlargement of hepatic sinuses around portal vein, vacuolar degeneration of hepatocytes and scattered nuclear fragmentation in PV3 group. Visceral enlargement, a large number of flaky gray-yellow necrosis foci appeared on the surface. HE staining showed that the hepatic sinuses around

the portal vein were obviously enlarged, a large number of degeneration and necrosis of hepatocytes were observed, and neutrophil infiltration was detected in the interstitium (Fig 1).

## Hypoxia was increased as indicated by pimonidazole staining

To investigate the hypoxia level in the liver, we did pimonidazole staining, a hypoxia indicator. The pimonidazole positive cells showed brown signaling in cytoplasm (Fig 2). After quantification, we found that in PV1 group, the hypoxia level was (+) on the 1st to 3rd and 7th day after operation. In PV2 group, the hypoxia level was (+)on the 1st to 2nd day after operation, and (++) on the 3rd day and (+) on the 7th day after operation. However, in PV3 group, the hypoxia level was (+++) on all of the time points we detected, which indicated that the most severity of hypoxia. All of the control group are negative (-)(Fig 2). These results showed that the more amount of microcapsules transplanted, the /more severity of hypoxia was.

## $\Delta R2^*$ is decreased with the amount of microcapsules transplanted

The BOLD signal increased after carbogen gas administration (Fig 3). The $\Delta R2^*$ value in each group was shown in Table 1 and Fig 4. With the increase of portal vein microcapsule transplantation dosage, the$\Delta R2^*$ value decreased gradually. Within 3 days after operation, the$\Delta R2^*$ value decreased with the prolongation of time, and increased on the 7th day after operation. When the dosage of portal vein transplantation was 5000 microbeads/kg, the$\Delta R2^*$ was the smallest, indicating the strongest effect. The statistical analysis was shown in Fig 4.

## The correlation of the $\Delta R2^*$ with the amount of transplanted microcapsules and hypoxia level

We noticed that after 3 days of transplantation, the hypoxia level increased severely, then we detected the correlation of $\Delta R2^*$ with hypoxia level and the amount of transplanted microcapsules on this timepoint. On the 3rd day after transplantation, the$\Delta R2^*$ value decreased with the increase of transplantation dose. Whilst, we also detected the most strong signaling level with pimonidazole staining. Nonparametric Spearman test showed that the$\Delta R2^*$ was negatively correlated with transplantation dose (r = -0.932, P<0.001, Fig 5A) and pimonidazole signaling (r = -0.893, $p$<0.001, Fig 5B).

## Discussion

Islet cell transplantation is an ideal method for the treatment of type I diabetes mellitus, which is simple and convenient with low adverse reactions. It has been widely used, but it faces severe immunological rejection after operation. In 1980, Lim and Sun initiated the transplantation of rat islet cells encapsulated in alginate-polylysine microcapsules [21]. It was a translucent membrane that allowed oxygen, electrolyte, small molecule nutrients and metabolites to pass through but keep the islet cells unaffected by the immune system, and provided an ideal method for solving the immunological rejection of islet cell transplantation.

Previous study found that vascular regeneration after islet cell transplantation is the key factor to ensure the survival and function of islets [22]. However, islet revascularization takes 10–14 days, and one of the important factors affecting islet survival is hypoxia [23–24]. In the early stage of liver microencapsulated islet transplantation, the oxygen consumed by islet cells are mainly supplied by diffusion at the transplantation site. Staying in small branches of portal vein with large doses of microcapsules may cause vascular embolism, leading to ischemia in surrounding liver tissue and then seriously affecting the survival of islet cells. Therefore, the early stage after transplantation is an important factor affecting the success or failure of

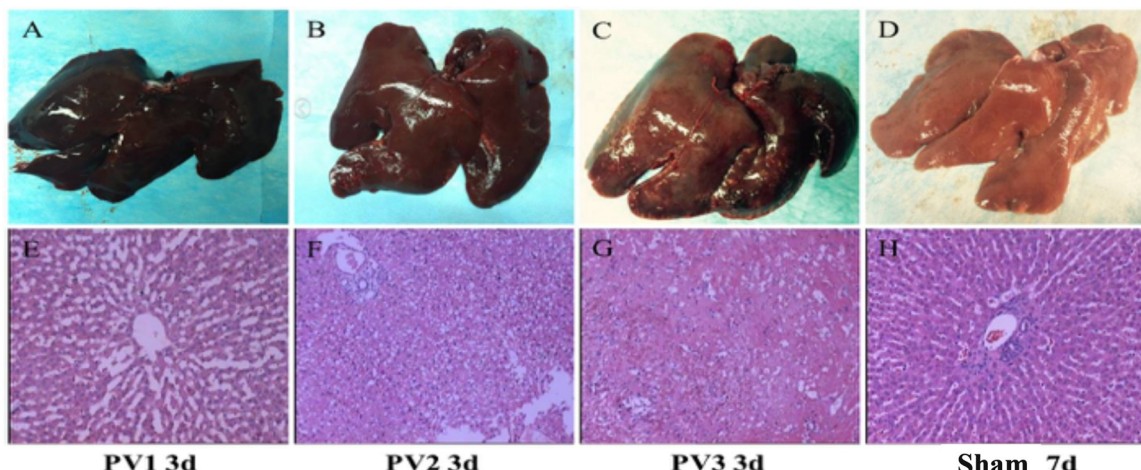

**Fig 1. Morphological observation and Hematoxylin-eosin staining of liver in each group on the 3rd day after transplantation.** (A) and (E), (B) and (F), (C) and (G) were pathological and HE staining results on the 3rd day after transplantation in PV1, PV2 and PV3 groups; (D) and (H) was pathological and HE staining results on the 7th day after transplantation in the control group. In PV1 group, hepatic sinusoids enlarged slightly; in PV2 group, a small amount of hepatocytes degenerated and necrotized; in PV3 group, a large number of hepatocytes degenerated and necrotized; in control group, there was no abnormal change in hepatocytes.

transplantation. For the liver function damage caused by hypoxia after transplantation, several

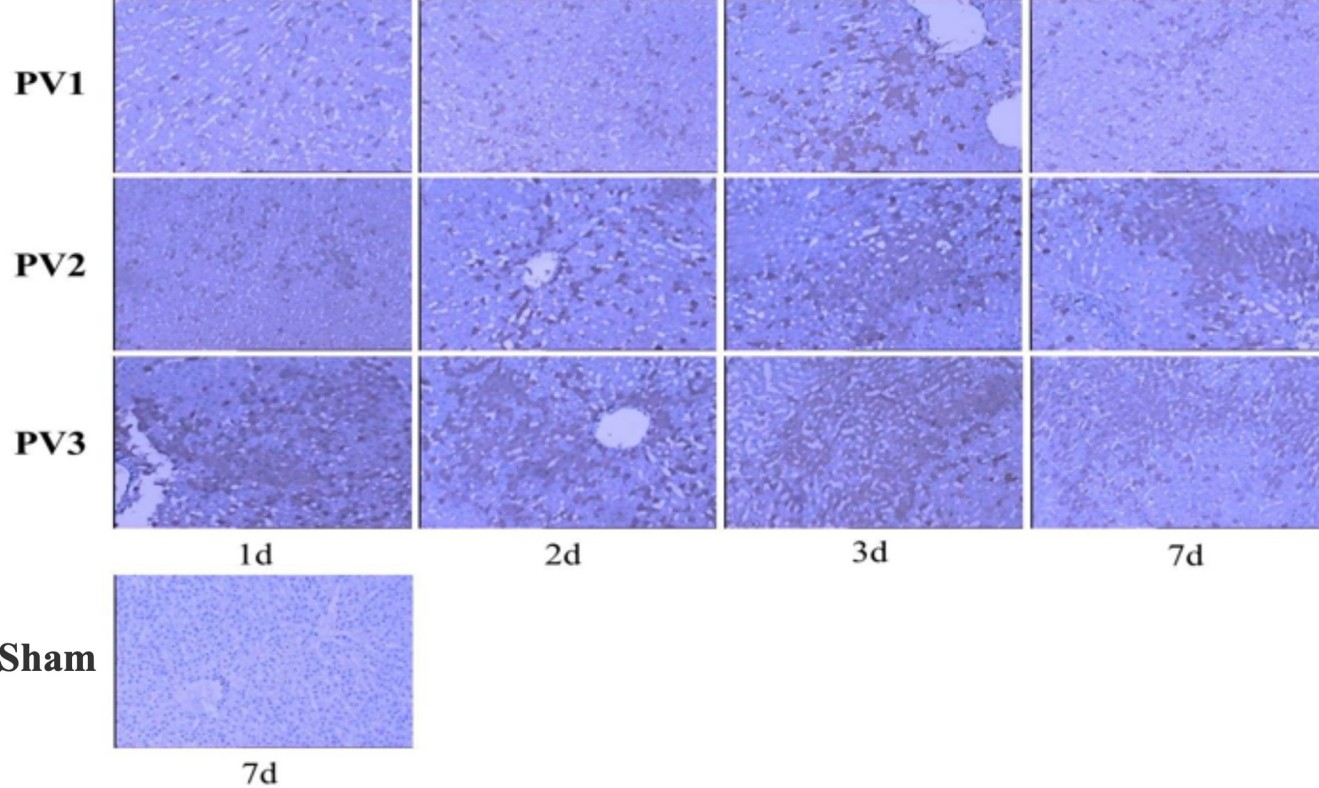

**Fig 2. Immunohistochemical staining of pimonidazole in each group.** In PV1 group, the hypoxia level was (+) on the 1st to 3rd and 7th day after operation. In PV2 group, the hypoxia level was (+)on the 1st to 2nd day after operation, and (++) on the 3rd day and (+) on the 7th day after operation. In PV3 group, the hypoxia level was (+++) on all of the time points.

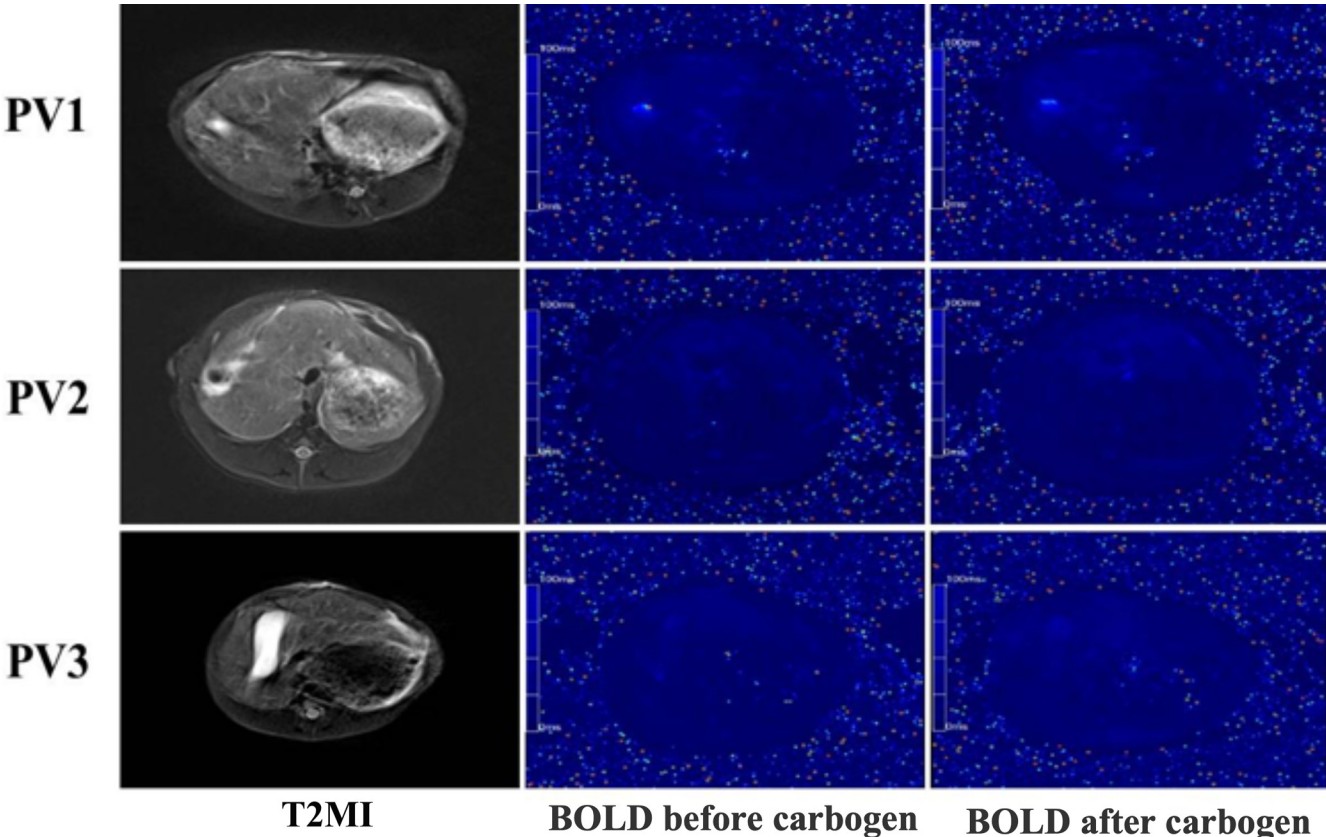

**Fig 3. BOLD-fMRI parametric maps.** The BOLD signal increased after carbogen gas administration. With the increase of portal vein microcapsule transplantation dosage, theΔR2* value decreased gradually.

rounds of transplantation are often used clinically to reduce this damage, which makes it difficult to monitor the amount of transplantation. Thus, it is necessary to setup an effective in vivo monitoring method. Sakada et al. found that the most severe liver injury occurred on the second day after islet cell portal vein transplantation, meanwhile after one month of transplantation, liver damage was less severe than that at an early stage [25]. Davalli et al. found that the number of islet cells and insulin content decreased significantly on the first to third day after transplantation. Besides of instant blood mediated inflammatory reaction at the early stage of transplantation, the ischemia-hypoxia mediated injury of recipient liver tissue is an important factor leading to the survival of islet cells [26]. In this study, we monitored hepatic hypoxia on the 1st, 2nd, 3rd and 7th day after operation, and pimonidazole immunohistochemical results were used as a reference for hypoxia. Pimonidazole is a hypoxic probe that can bind to hypoxic cells and identify them with antibodies [27–29]. This study found that the degeneration and

**Table 1. The ΔR2* value at different time points in each group (Hz, x̄±s).**

| Time | 1d | 2d | 3d | 7d |
|------|------|------|------|------|
| PV1 | 20.32±2.31 | 20.00±2.60 | 19.45±2.01 | 20.27±2.31 |
| PV2 | 16.78±2.34 | 15.81±1.69 | 14.99±1.40 | 16.10±1.40 |
| PV3 | 10.30±1.67 | 8.51±2.15 | 8.70±2.00 | 9.21±2.34 |
| Sham | 26.03±1.42 | 24.07±1.54 | 26.07±2.06 | 25.90±2.11 |

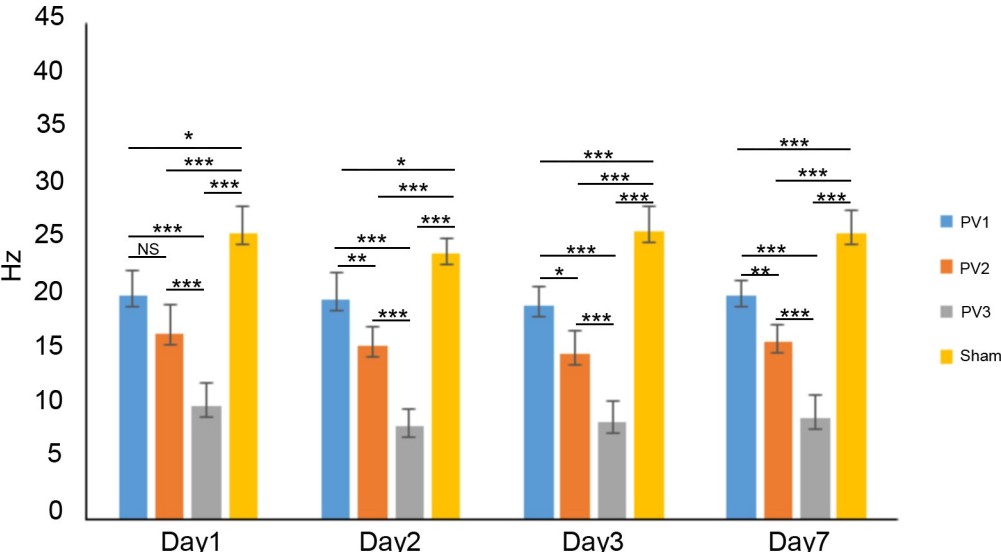

**Fig 4. Statistical analysis of the ΔR2* value in each group.** The ΔR2* values in each group after microcapsules transplanted on the 1st, 2nd, 3rd and 7th day were shown. Unpaired Student's two tailed t-Test were used for statistical analysis, with significance levels * for P<0.05, ** for P<0.01 and *** for P<0.001.

necrosis of hepatocytes were the most serious in PV3 group on the third day. When the dose of microcapsule transplantation increased gradually, the degree of liver hypoxia increased gradually. After microcapsule transplantation, the liver hypoxia became worse gradually. On the third day, it showed the most serious stage. On the seventh day, due to the strong compensatory function of the liver, hypoxia was alleviated, which was similar to the observation of Sakada et al. [25].

BOLD-fMRI uses deoxyhemoglobin as an endogenous contrast agent to observe the changes of signal to understand the local tissue oxygen saturation and tissue oxygen content, thus reflecting the changes of tissue metabolism, hemodynamics and function. R2* value is

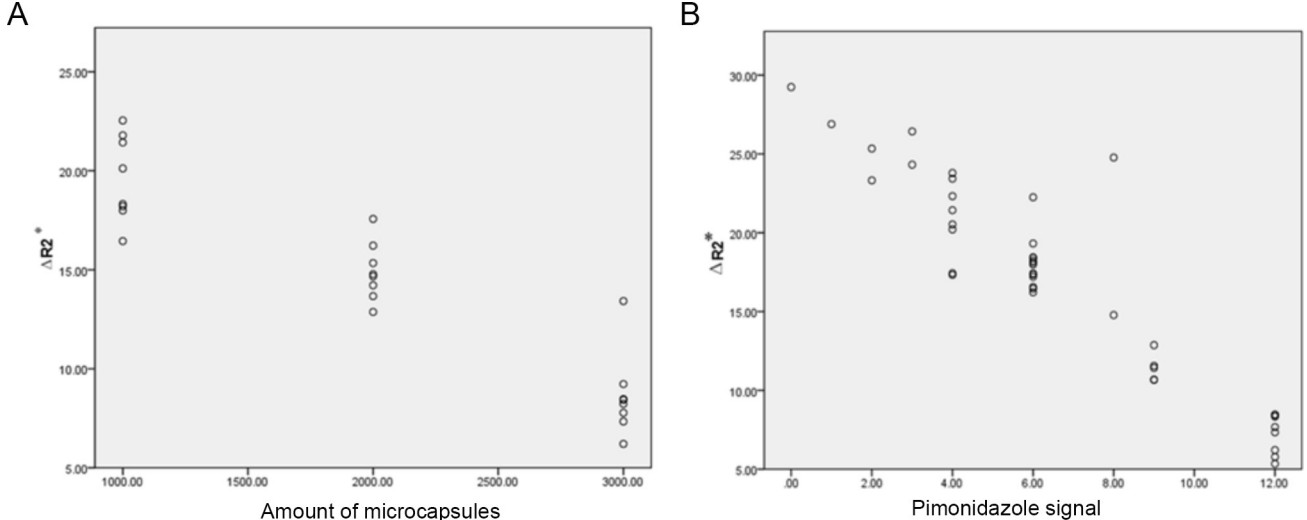

**Fig 5. The correlation of the ΔR2* with the amount of transplanted microcapsules and hypoxia level.** The ΔR2* value was negatively correlated with the transplantation dose (r = -0.932, P<0.001, Fig 5A) and pimonidazole signaling (r = -0.893, p<0.001, Fig 5B). on the 3rd day after transplantation.

negatively correlated with tissue oxygen content. The higher $R2^*$ value, the more deoxyhemoglobin, the lower oxygen partial pressure and the lower signal on BOLD image. Previous studies have shown that the changes of hepatic hemodynamics and venous blood flow can be understood by observing the changes of BOLD signal intensity under different conditions of oxygen stimulation [30], so as to further understand the changes of hepatic vascular structure and function. Other studies have used BOLD imaging to evaluate hepatic fibrosis [17,31–32]. Compared with normal liver, the response of fibrotic liver to Carbogen stimulation decreases due to the change of hemodynamics. It shows that the $R2^*$ value decreases before and after stimulation, and with the aggravation of fibrosis, the $\Delta R2^*$ value decreases. Carbogen-mediated functional magnetic resonance imaging of liver is very sensitive to changes in hepatic hemodynamics. The less portal venous blood flow, the less response to Carbogen gas stimulation. In this study, we used the classical Carbogen stimulation method, and to reduce the influence of abdominal breathing on the image and the deviation of the $\Delta R2^*$ value, we applied pressure on the abdomen with sand bags and controlled the breathing gating. The results showed that BOLD-fMRI were consistent with the pathological results. The $\Delta R2^*$ value in PV1, PV2 and PV3 groups decreased significantly compared with the control group. This may be due to the increase of the dose of microcapsules. Portal venous blood flow decreased gradually, and the response of liver to Carbogen stimulation decreased gradually as indicated by the $\Delta R2^*$. We also found that the $\Delta R2^*$ value decreased within 1, 2 and 3 days after operation, but increased on the 7th day. This may be due to the fact that the early compensation mechanism after microcapsule transplantation can not compensate for the decrease of liver blood flow, but it can still have a buffering effect. The degree of change of hepatic parenchymal deoxyhemoglobin concentration is relatively reduced, thereby narrowing the difference between groups. In this study, we observed a significant negative correlation between the $\Delta R2^*$ value and the amount of transplantation. Therefore, BOLD-fMRI stimulated by Carbogen can assess the degree of liver hypoxia after portal vein microcapsule transplantation, providing a new means for early clinical intervention and the amount of transplantation tolerated by liver.

In conclusion, BOLD-fMRI stimulated by Carbogen can evaluate the degree of liver hypoxia after portal vein microcapsule transplantation, and provide a new means for early clinical intervention and guidance of liver tolerance.

## Author Contributions

**Conceptualization:** Yuefu Zhan, Jianqiang Chen.

**Formal analysis:** Yuefu Zhan.

**Funding acquisition:** Jianqiang Chen.

**Investigation:** Yuefu Zhan, Yehua Wu.

**Methodology:** Yuefu Zhan, Yehua Wu, Jianqiang Chen.

**Writing – original draft:** Jianqiang Chen.

**Writing – review & editing:** Jianqiang Chen.

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
