## [Decision Letter · Decision Letter 0]

29 Aug 2019

PONE-D-19-17327

Carbogen gas-challenge BOLD fMRI in assessment of liver hypoxia after portal microcapsules implantation

PLOS ONE

Dear Dr. Chen,

Thank you for submitting your manuscript to PLOS ONE. After careful consideration, we feel that it has merit but does not fully meet PLOS ONE’s publication criteria as it currently stands. Therefore, we invite you to submit a revised version of the manuscript that addresses the points raised during the review process.

The manuscript has been reviewed by two experts in the field. Both felt that the use of carbogen BOLD to evaluate hepatic hypoxia after transplantation was novel and interesting. However, they differed on the quality of the experimental design and results presented. Both reviewers commented that more complete statistical analysis is needed. Reviewer #2 raised a number of further issues, including serious concerns about the experimental design and how the results are presented. If these critiques can be addressed, then a major revision of the manuscript should be able to address the remaining, more minor comments. Please also have the manuscript edited for clarity and grammar with the help of a native English speaker.

We would appreciate receiving your revised manuscript by Oct 13 2019 11:59PM. To enhance the reproducibility of your results, we recommend that if applicable you deposit your laboratory protocols in protocols.io, where a protocol can be assigned its own identifier (DOI) such that it can be cited independently in the future. For instructions see: http://journals.plos.org/plosone/s/submission-guidelines#loc-laboratory-protocols

We look forward to receiving your revised manuscript.

Kind regards,

Nick Todd, PhD

Academic Editor

PLOS ONE

Journal Requirements:

"We are grateful to the colleagues in our group and hospital for their suggestions and supports. This work was supported by Hainan Province Scientific and Technological Program (No. ZDYD2019164); Health Family Planning Project in Hainan Province (16A200087)."

Please provide an amended Funding Statement that declares *all* the funding or sources of support received during this specific study (whether external or internal to your organization) as detailed online in our guide for authors at http://journals.plos.org/plosone/s/submit-now

Please state what role the funders took in the study.  If any authors received a salary from any of your funders, please state which authors and which funder. If the funders had no role, please state: "The funders had no role in study design, data collection and analysis, decision to publish, or preparation of the manuscript."

Reviewers' comments:

Reviewer's Responses to Questions

**Comments to the Author**

1. Is the manuscript technically sound, and do the data support the conclusions?

Reviewer #1: Yes

Reviewer #2: No

2. Has the statistical analysis been performed appropriately and rigorously? 

Reviewer #1: Yes

Reviewer #2: No

3. Have the authors made all data underlying the findings in their manuscript fully available?

Reviewer #1: Yes

Reviewer #2: No

4. Is the manuscript presented in an intelligible fashion and written in standard English?

Reviewer #1: Yes

Reviewer #2: No

5. Review Comments to the Author

Reviewer #1: 1. Introduction: The Purpose of using Carbogen gas and the meaning of gas-challenge in the title need to be described clearly.

2. How long did it take to perform BOLD-fMRI data acquisition?

3. Results: The authors need to provide the tables to list the ΔR2*value of BOLD-fMRI in each group，and show the statistical results。

4. There are no pathological or imaging images.

5. In Table 1, if adding the P value will make the results clearer.

6. The full name of the BOLD is blood oxygen level dependent。Similar grammatical mistakes and clerical error may exist in several sentences in the manuscript，for example page 16, the first sentence“it still has buffering effect and liver parenchymal deoxyhemoglobin concentration”。Please carefully edit the language issues with the help of the native English-speaking experts.

Reviewer #2: Introduction:

1. It is too simple for the intro section。The author mentioned hypoxia or related concept in the intro section repeatedly，please identity what exactly the mechanism of hypoxia happening after Islet cell transplantation and the concept of buffering effect by artery and portal vein in liver should be discussed in the process of hypoxia

2. The mechanism of BOLD should be expounded in the intro section。

3. In“By observing the changes of T2WI signal”，should T2WI be written as T2*？

4. Please explain why you choose the Carbogen gas as the stimulant for BOLD instead of pure oxygen，especially the effect of 5% CO2

5. Please identify what ΔR2* is。

Materials and methods

1. Please identify what SPF condition is

2. Please identify the unit of transplanted microcapsules in the 1000/kg, 3000/kg and 5000/kg

3. I can not find the information in this manuscript whether the islet cell was encapsulated into the microcapsules or not，which means nothing encapsulated in the study。If the effect to liver hypoxia by microcapsules is different between islet cell encapsulated and not，then it was the major deficiency to this study，or author should prove that there is no difference in affecting liver hypoxia between islet cell encapsulated and not。We do not use the empty microcapsules for treatment after all。

4. It is better to replace saline with the solution of microcapsules（not included microcapsules） for the control group

5. I wonder why MR scan was not performed before microcapsules transplant。It is very important to get the baseline for observing changes before and after operation。Data and experiment need to be supplemented。

6. There were 51 rabbits included but only 41 were used（three for 1,2,3,7 day in three experimental groups and 6 for control group），please explain

7. “The BOLD scan was performed at a flow rate of 15 ml/s.”，it meant the flow rate in the mask？

8. The author should clarify the amount of data for each group。As my understanding to this manuscript，author used 3*5 ROIs（15）for each group to meet the sample size requirement for statistical analysis and the author should talked about the “type I error” because there were only 3 rabbits in each group after all。Inversely，If the average value from 5 ROIs was used，the date is not enough for statistical analysis at all。

9. The author should address more details about the histopathological examination ，like the sections for whole liver or partial？like how to choose the section which is used for microscopy analysis。I do not think “Five ROIs were randomly selected”is appropriate for the inhomogeneity of hypoxia which should be talked in detail。In the end，the ROIs used in BOLD analysis and histopathological analysis should be in the same/similar location which make this study more credible。

10. In the last paragraph，should R2* be written as ΔR*？

RESULTS

1. Statistical analysis between BOLD and histopathological examinations are much needed （like some quantitative indicators for avascular necrosis or the amount of capillaries，or the pimonidazole results，et al）to guarantee the value and reliability of this study。In this study，one key evidence was missed in the relationship between the amount of used microcapsules and the ΔR2*，in which the essence of hypoxia is ischemia result from obstruction of microcapsules in small vessels，hence the result from ischemia should be quantified and compared to BOLD。I only find the descriptive words for pimonidazole results but no statistical analysis。

2. In “2.1 Histopathological examination”，please indicate the time for each observation

3. Please identify the meaning of “+”for pimonidazole results

4. In “2.3：but there was significant difference between each two groups (P < 0.05).” meant significant difference between each two groups in each day？

5. In“2.4”，are all the result from 3rd day？Please address the statistical relationship between transplantation dose and ΔR2* in other time point rather than only 3rd day

Discussion

1. In the 2rd paragraph，“and pimonidazole immunohistochemical results were used as a reference for hypoxia”is improper because there was no statistical analysis between and BOLD or transplantation dose

2. In“and the response of liver to Carbogen gas stimulation decreased gradually”，please identify what the response of liver is

3. There are many times for miswriting ΔR2* as R2*，the author should check it carefully。

4. Please identify “compensatory mechanism”in detail and it is hard to understand“it still has buffering effect and liver parenchymal deoxyhemoglobin concentration”，please make it more readable

5. Please identify what change is in“The degree of change was relatively reduced”

6. It is very sloppy to make the conclusion ：“Therefore, BOLD-fMRI stimulated by Carbogen gas can assess the degree of liver hypoxia after portal vein microcapsule transplantation”due to the key evidence was missed which can be fixed by doing statistical analysis related to pimonidazole results or some quantitative indicators for avascular necrosis or the amount of capillaries

7. It is very hard to understand the last sentence “It is worthwhile……”，please rewrite it for more readability

Figure legends:

The fig4 legend should indicate the time

6. PLOS authors have the option to publish the peer review history of their article (what does this mean?). If published, this will include your full peer review and any attached files.

Reviewer #1: No

Reviewer #2: No

---

## [Author Response · Author response to Decision Letter 0]

15 Oct 2019

Response to Reviewers：

Reviewer #1: 1. Introduction: The Purpose of using Carbogen gas and the meaning of gas-challenge in the title need to be described clearly.

RE：Thank you very much. We revised our manuscript according to your nice suggestion. 

2. How long did it take to perform BOLD-fMRI data acquisition?

RE：Normally one scan takes less than 10min. However, the animals needs to be treated before MRI scan. Totally, the whole procedure needs around 30min for one animal. 

3. Results: The authors need to provide the tables to list the ΔR2*value of BOLD-fMRI in each group，and show the statistical results。

RE：Thank you for the suggestion, we provided a new figure in our revised manuscript. 

4. There are no pathological or imaging images.

RE：Yes, we had those images, they were shown in figure 1, Thanks. 

5. In Table 1, if adding the P value will make the results clearer.

RE：Thank you, as you suggested in the second question, we provide a new figure with statistical results. 

6. The full name of the BOLD is blood oxygen level dependent。Similar grammatical mistakes and clerical error may exist in several sentences in the manuscript，for example page 16, the first sentence“it still has buffering effect and liver parenchymal deoxyhemoglobin concentration”。Please carefully edit the language issues with the help of the native English-speaking experts.

RE：Thank you very much. We modified the manuscript carefully and corrected some typos and mistakes. 

Reviewer #2: Introduction:

1. It is too simple for the intro section. The author mentioned hypoxia or related concept in the intro section repeatedly，please identity what exactly the mechanism of hypoxia happening after Islet cell transplantation and the concept of buffering effect by artery and portal vein in liver should be discussed in the process of hypoxia

2. The mechanism of BOLD should be expounded in the intro section。

RE: for 1 and 2: Thank you so much, we modified our introduction and discussion. The questions raised by the reviewer had been addressed. 

3. In“By observing the changes of T2WI signal”，should T2WI be written as T2*？

RE： Thank you for your suggestion, we corrected T2WI as T2*. 

4. Please explain why you choose the Carbogen gas as the stimulant for BOLD instead of pure oxygen，especially the effect of 5% CO2 

RE：Thanks for your question. Carbogen (95% O2 and 5% CO2) is widely used in experimental and clinical studies. Carbogen administration is known to be safe in humans and has been clinically applied during radiation therapy in order to raise the oxygen tension and increase the radio-sensitivity of the anoxic region. Compared with using pure oxygen, incorporating 5% CO2 is believed to counteract any oxygen-induced vasoconstriction.

5. Please identify what ΔR2* is。

RE: Thanks for your question. ΔR2* value = R2* air-R2* Carbogen gas, which was present in “1.4 Image Analysis and Post-processing”. 

Materials and methods

1. Please identify what SPF condition is

RE: The SPF is specific pathogen free, we added the full name in the new manuscript. 

2. Please identify the unit of transplanted microcapsules in the 1000/kg, 3000/kg and 5000/kg 

RE: Thank you. The unit of microcapsules is the absolute number detected under the stereotype microscope. We added a paragraph of describe this in the method section. Thanks again. 

3. I can not find the information in this manuscript whether the islet cell was encapsulated into the microcapsules or not，which means nothing encapsulated in the study。If the effect to liver hypoxia by microcapsules is different between islet cell encapsulated and not，then it was the major deficiency to this study，or author should prove that there is no difference in affecting liver hypoxia between islet cell encapsulated and not。We do not use the empty microcapsules for treatment after all。

RE: Thanks for your nice comments. Yes, you are right. We encapsulated nothing in the experiment. 

In this study, what we want to provide is a convenient way to detect the oxygen level when microcapsules were used in the treatment for diabetes. No matter whether the islet cell were encapsulated, the hypoxia always happened. However, relationship of the hypoxia level and the number of microcapsules transplanted was not well studied. So in the study, we detected their correlation and also tested that BOLD-fMRI is an easy and sensitive way to estimate the level of hypoxia. 

4. It is better to replace saline with the solution of microcapsules（not included microcapsules） for the control group 

RE: Thank you for your suggestion. Actually, we still believe saline is a better control group, because we wash and resuspend the microcapsules in saline when it was implanted. 

5. I wonder why MR scan was not performed before microcapsules transplant。It is very important to get the baseline for observing changes before and after operation。Data and experiment need to be supplemented。

RE: Thank you very much, in our experiments, in each MRI scan we had a control group as the baseline which is reasonable and quite widely used by this kind of studies. 

6. There were 51 rabbits included but only 41 were used（three for 1,2,3,7 day in three experimental groups and 6 for control group），please explain

RE: Thank you for this nice question. Yes, at the beginning we prepared 15 rabbits for each group in case of unexpected dying during the surgery or after surgery. However, we were so lucky, those animals were in good conditions during and after the surgeries. Therefore, on the basis of “3R rules” (Reduction, Replacement and Refinement), we did not sacrifice all of the animals. 

7. “The BOLD scan was performed at a flow rate of 15 ml/s.”，it meant the flow rate in the mask？

RE: Yes, 15ml/s means the flow rate of air and Carbogen in the mask.

8. The author should clarify the amount of data for each group。As my understanding to this manuscript，author used 3*5 ROIs（15）for each group to meet the sample size requirement for statistical analysis and the author should talked about the “type I error” because there were only 3 rabbits in each group after all。Inversely，If the average value from 5 ROIs was used，the date is not enough for statistical analysis at all。

RE: Thank you for your question. Yes, in this study we used 3 animals to meet the sample size. In the analysis of images, we chose the same region (ROI) before and after carbongen-challenge. Five ROIs were selected in each animal to calculate the ΔR2* value. In this system, 3 animals is sufficient to undergo statistical analysis. 

9. The author should address more details about the histopathological examination ，like the sections for whole liver or partial？like how to choose the section which is used for microscopy analysis。I do not think “Five ROIs were randomly selected”is appropriate for the inhomogeneity of hypoxia which should be talked in detail。In the end，the ROIs used in BOLD analysis and histopathological analysis should be in the same/similar location which make this study more credible。

RE: Thanks for the nice suggestion. Yes, we used the same location for the histopathological examination with BOLD analysis, which was added in the new manuscript. Then, five ROIs were randomly selected in the location. Thanks again for your suggestion. 

10. In the last paragraph，should R2* be written as ΔR*？

RE: Thank you very much. Yes, it is ΔR*, we are sorry for the carelessness. 

RESULTS

1. Statistical analysis between BOLD and histopathological examinations are much needed （like some quantitative indicators for avascular necrosis or the amount of capillaries，or the pimonidazole results，et al）to guarantee the value and reliability of this study。In this study，one key evidence was missed in the relationship between the amount of used microcapsules and the ΔR2*，in which the essence of hypoxia is ischemia result from obstruction of microcapsules in small vessels，hence the result from ischemia should be quantified and compared to BOLD。I only find the descriptive words for pimonidazole results but no statistical analysis。

RE：Thank you for the wonder suggestions. A new figure was incorporated in the new manuscript, which showed the rΔR2* value in different groups at distinct timepoint. Now the results were much clearer. we can notice a negative corrlation between the amount of microcapsules transplanted and the ΔR2* value. Additionally, In this figure, the statistical data were also incorporated. 

With regards to the histopathological examinations, honestly, we did not observe apparent obstruction of microcapsules in the vessels, which is also not easy to identify in clinical.

However, with the Pimondazole staining we can observe apparent difference with different amount of microcapsules and the severity was increased with time passed by. For the statistical analysis, we calculated the hypoxia level of each ROI based on the method used by Nordsmark, The difference was so dramatic. Additionally, we also plotted a new figure to show the correlation between ΔR2* and hypoxia level (Fig. 5B). As the degree of hypoxia in the liver gradually increased, the average value of ΔR2* decreased. Using the nonparametric Spearman test, the average value of ΔR2* was highly negatively correlated with the hypoxic marker, and the correlation coefficient was (r=-0.893, p<0.001).

2. In “2.1 Histopathological examination”，please indicate the time for each observation

RE: Thanks a lot, we added the timepoint for each observation. 

3. Please identify the meaning of “+”for pimonidazole results

RE: Thanks for your question, we added a paragraph to describe how to calculate the results based on the widely used method, and the meaning of “-“, “+”, “++”, “+++”. 

4. In “2.3：but there was significant difference between each two groups (P < 0.05).” meant significant difference between each two groups in each day？

RE: Yes, it is. To make the point clear, we plotted a new figure 4 to show the statistic result. 

5. In“2.4”，are all the result from 3rd day？Please address the statistical relationship between transplantation dose and ΔR2* in other time point rather than only 3rd day

RE: Thanks a lot for this suggestion. A new figure 4 was incorporated in the new manuscript, we can notice a clear correlation of transplantation dose and ΔR2*. 

Discussion

1. In the 2rd paragraph，“and pimonidazole immunohistochemical results were used as a reference for hypoxia”is improper because there was no statistical analysis between and BOLD or transplantation dose.

RE: Thanks for your nice suggestions. According your questions, we added and replaced several figures, we believe those new data can support our conclusion. 

2. In“and the response of liver to Carbogen gas stimulation decreased gradually”，please identify what the response of liver is

RE: Thank you very much, the responds was indicated by ΔR2*, we modified it in the manuscript. 

3. There are many times for miswriting ΔR2* as R2*，the author should check it carefully。

RE: Thank you. We are so sorry for the carelessness. 

4. Please identify “compensatory mechanism”in detail and it is hard to understand“it still has buffering effect and liver parenchymal deoxyhemoglobin concentration”，please make it more readable

RE: Thanks, we modified it and made it more readable. 

5. Please identify what change is in “The degree of change was relatively reduced”

RE: Thank you. We correct our manuscript carefully. 

6. It is very sloppy to make the conclusion ：“Therefore, BOLD-fMRI stimulated by Carbogen gas can assess the degree of liver hypoxia after portal vein microcapsule transplantation”due to the key evidence was missed which can be fixed by doing statistical analysis related to pimonidazole results or some quantitative indicators for avascular necrosis or the amount of capillaries

RE: Thank you. According your nice suggestions, we modified our manuscript and provided the statistical analysis results and some new data. 

7. It is very hard to understand the last sentence “It is worthwhile……”，please rewrite it for more readability

RE: we modified the last paragraph to make it readable. 

Figure legends:

The fig4 legend should indicate the time

RE: Thank you. We corrected it.

---

## [Decision Letter · Decision Letter 1]

11 Nov 2019

Carbogen gas-challenge BOLD fMRI in assessment of liver hypoxia after portal microcapsules implantation

PONE-D-19-17327R1

Dear Dr. Chen,

We are pleased to inform you that your manuscript has been judged scientifically suitable for publication and will be formally accepted for publication once it complies with all outstanding technical requirements.

With kind regards,

Nick Todd, PhD

Academic Editor

PLOS ONE

Additional Editor Comments (optional):

Reviewers' comments:

Reviewer's Responses to Questions

**Comments to the Author**

1. If the authors have adequately addressed your comments raised in a previous round of review and you feel that this manuscript is now acceptable for publication, you may indicate that here to bypass the “Comments to the Author” section, enter your conflict of interest statement in the “Confidential to Editor” section, and submit your "Accept" recommendation.

Reviewer #1: All comments have been addressed

2. Is the manuscript technically sound, and do the data support the conclusions?

Reviewer #1: Yes

3. Has the statistical analysis been performed appropriately and rigorously? 

Reviewer #1: Yes

4. Have the authors made all data underlying the findings in their manuscript fully available?

Reviewer #1: Yes

5. Is the manuscript presented in an intelligible fashion and written in standard English?

Reviewer #1: Yes

6. Review Comments to the Author

Reviewer #1: The author has sincerely ansered the auestions.I feel that this manuscript is now acceptable for publication.

7. PLOS authors have the option to publish the peer review history of their article (what does this mean?). If published, this will include your full peer review and any attached files.

Reviewer #1: No

---

## [Editor Report · Acceptance letter]

18 Nov 2019

PONE-D-19-17327R1 

Carbogen gas-challenge BOLD fMRI in assessment of liver hypoxia after portal microcapsules implantation 

Dear Dr. Chen:

I am pleased to inform you that your manuscript has been deemed suitable for publication in PLOS ONE. Congratulations! Your manuscript is now with our production department. 

With kind regards,

on behalf of

Dr. Nick Todd 

Academic Editor

PLOS ONE